# Enhancing the Toughness of PAA/LCNF/SA Hydrogel through Double-Network Crosslinking for Strain Sensor Application

**DOI:** 10.3390/polym16010102

**Published:** 2023-12-28

**Authors:** Xin Li, Hui Gao, Qiang Wang, Shanshan Liu

**Affiliations:** 1State Key Laboratory of Biobased Material and Green Papermaking, Qilu University of Technology, Shandong Academy of Sciences, Jinan 250353, Chinaliushanshan@qlu.edu.cn (S.L.); 2Key Laboratory of Paper Science and Technology of Ministry of Education, Faculty of Light Industry, Qilu University of Technology, Shandong Academy of Sciences, Jinan 250353, China

**Keywords:** sodium alginate, polyacrylic acid, double-network hydrogel, lignocellulose nanofibril, flexible sensor

## Abstract

Lignin-containing nanocellulose fibers (LCNF) have been considered as a valuable enhancer for polyacrylic acid (PAA)-based hydrogels that can form rigid porous network structures and provide abundant polar groups. However, the PAA–LCNF hydrogel is dominated by a single-network (SN) structure, which shows certain limitations when encountering external environments with high loads and large deformations. In this paper, sodium alginate (SA) was introduced into the PAA–LCNF hydrogel network to prepare a double-network (DN) hydrogel structure of the SA-Ca^2+^ and PAA–LCNF through a two-step process. The covalent network of PAA–LCNF acts as the resilient framework of the hydrogel, while the calcium bridging networks of SA, along with the robust hydrogen bonding network within the system, function as sacrificial bonds that dissipate energy and facilitate stress transfer. The resulting hydrogel has porous morphologies. Results show that SA can effectively improve the mechanical properties of DN hydrogels and endow them with excellent thermal stability and electrical conductivity. Compared with pure PAA–LCNF hydrogel, the elongation at break of DN hydrogel increased from 3466% to 5607%. The good electrical conductivity makes it possible to use the flexible sensors based on DN hydrogel to measure electrophysiological signals. Our results can provide a reference for developing multifunctional hydrogels that can withstand ultra large deformation.

## 1. Introduction

Toughness is one of the most important properties of polymer hydrogels [1], which could prevent hydrogels from cracking or damaging when being subjected to cycled mechanical deformation, including compression, stretching, or twisting [2]. This is essential to ensure the long-term stability of hydrogels in applications where the capability of withstanding mechanical stress or deformation is required [3,4,5,6]. Modifying the network structures and energy dissipation patterns are current strategies to enhance the toughness and mechanical properties of hydrogels [7,8,9,10]. The main approaches to change the network include increasing the crosslinking density of the hydrogel through crosslinking regulation [11], adjusting the composition of the polymer, adding additives and crosslinking agents, and incorporating other materials like nanoparticles, cellulose, etc. [12,13,14,15]. By strategically altering the network architecture, researchers aim to provide hydrogels with enhanced mechanical properties, paving the way for their effective use in a variety of applications requiring durability and resistance to mechanical challenges. These innovative strategies are helping to advance the field of hydrogel design, ensuring their suitability for diverse and challenging real-world applications.

In recent years, lignin-containing nanocellulose fibers (LCNF) have been considered as a valuable enhancer for PAA-based hydrogels due to their advantages like low cost, environmental friendliness, excellent biocompatibility, and thermal stability. On one hand, lignin in LCNF can act as a spacer to form a rigid porous network structure during the hydrogel formation process [16]. On the other hand, the surface of LCNF is rich in polar hydroxyl and carboxyl groups, which can interact with the polymer matrix of the hydrogel through hydrogen bonds. This interaction facilitates energy dissipation, thereby endowing the hydrogel with enhanced mechanical strength. However, the hydrogel system constructed with LCNF is dominated by a single-network structure, which shows certain limitations when encountering external environments with high loads and large deformations.

The double-network (DN) gels possess an interpenetrating polymer network (IPN) structure, wherein two networks exhibit distinct properties, including network density, stiffness, molecular weight, crosslinking density, and more, showcasing sharp contrasts between the two interconnected networks. It is a promising strategy that can significantly enhance the toughness of hydrogels through careful design and regulation of two interpenetrating networks and component networks. In 2003, Gong et al. [17] introduced the concept of DN hydrogels by designing a chemically crosslinked poly-2-acrylamide-2-methylpropanesulfonic acid (PAMPS)/polyacrylamide (PAAm) hydrogel. The DN hydrogel was synthesized through a two-step continuous free radical polymerization process. The PAMPS component formed the first network, while the PAAm component formed the second network. This combination significantly enhanced the mechanical properties of the hydrogel. Gong et al. [18] further summarized the design principles for DN hydrogels as follows: the first network should consist of a brittle and rigid polymer, such as a polyelectrolyte, while the second network should consist of a soft and ductile polymer, such as a neutral polymer. The molar concentration of the second network should be 20–30 times higher than that of the first network. The first network should have a high degree of crosslinking, while the second network should have a low degree of crosslinking but a high molecular weight. These design principles allow the sacrificial bonds in the first network to effectively dissipate energy, while the second network reduces the elastic free energy density, thereby maintaining the integrity of the hydrogel during deformation. This principle is considered universal and has been observed in various tough materials, including polymers, metals, ceramics [19], and natural tissues [20]. In recent years, there has been significant interest in utilizing DN hydrogels as intelligent soft materials with highly adjustable mechanical properties. One particular area of focus has been the development of flexible strain sensors using DN hydrogels. For example, Li et al. [21] demonstrated the creation of a strain sensor based on DN hydrogels that possesses remarkable characteristics such as stretchability, 3D printability, self-healing capabilities, flexibility, and recoverability. This sensor combines an ion-crosslinked κ-carrageenan network with a covalently crosslinked polyacrylamide network, enabling effective monitoring and differentiation of various human motions. Another noteworthy contribution comes from Gao’s group, who engineered a physical crosslinked DN hydrogel using hydrophobic interactions and ion crosslinking. This resulting DN hydrogel serves as an antifatigue, self-recovering, and highly sensitive wearable strain sensor for monitoring human motions [22]. More recently, Zhou [23] and colleagues developed a DN-hydrogel-based strain sensor with exceptional stretchability. Their approach involved the combination of an ion-crosslinked agar network, a covalently crosslinked acrylic network, and a dynamic and reversible ion-crosslinked coordination between carboxyl groups and Fe ions. These advancements in DN-hydrogel-based strain sensors have sparked significant research and development interest, offering a wide range of possibilities for monitoring and distinguishing human movements. DN hydrogels inherently demonstrate exceptional mechanical properties, including high fracture stress (1–10 MPa), large tensile strain (1000–2000%), and high fracture energy (>103 J/m^2^), enabling their applications in wide fields [24,25,26,27,28,29].

Sodium alginate (SA) [30] is a linear natural polymer with the characteristics of non-toxicity, degradability, and good biocompatibility, and has been widely used in extensive fields [31,32,33]. In the presence of Ca^2+^, SA can crosslink with Ca^2+^ to form a stable network structure. However, the structure of SA hydrogel is loose, resulting in low mechanical strength. The incorporation of LCNF into DN hydrogels not only exploits the advantages of nanocellulose, but also addresses the limitations of single-network structures, demonstrating the potential for tailored hydrogel designs with superior mechanical performance.

In this paper, we developed a stretchable, electrically conductive, and thermally stable double-network hydrogel by introducing SA into a PAA–LCNF-based hydrogel with a specific ratio. The novelty of this study was that the covalent network of PAA–LCNF acts as the resilient framework of the hydrogel, while the calcium bridging networks of SA, along with the robust hydrogen bonding network within the system, function as sacrificial bonds that dissipate energy and facilitate stress transfer. It showed significantly better toughness than the single-bond-network hydrogel. A stable crosslinked network structure was formed between the two compounds through hydrogen bonding and physically crosslinking interactions. This result led to the formation of a more tightly connected dual network structure between the SA-Ca^2+^ and PAA–LCNF networks, which enhanced the elongation at break of the hydrogel. Additionally, the addition of SA also made the DN hydrogel electrically conductive. This novel development demonstrates the versatility of SA in enhancing the mechanical and electrical properties of hydrogels, broadening their potential applications in various fields, especially in the field of flexible and responsive sensor technologies.

## 2. Materials and Methods

### 2.1. Materials

LCNF with 18.18% lignin was prepared following the procedures in Ref. [34]. Acrylic acid (AA, 99%), sodium alginate (SA, AR), sodium dodecyl sulfate (SDS, 98.5%), lauryl methacrylate (LMA, 96%), tetramethyl ethylenediamine (TMEDA, 99%), and ammonium persulfate (APS, 98%) were obtained from Shanghai Aladdin Biochemical Technology Co., Ltd. (Shanghai, China). Sodium chloride (NaCl, AR) and calcium chloride (CaCl_2_, AR) were purchased from Guomao Group Chemical Reagent Co., Ltd. (Changzhou, China). All materials in this study were used without further purification. Deionized water was used in all experiments.

### 2.2. Synthesis Procedures

The DN hydrogel was prepared by a two-step method, as shown in Figure 1. First, 0.35 g SDS, 0.26 g NaCl, and different amounts of SA (mass fraction of 0.2 wt%, 0.3 wt%, 0.4 wt%, 0.5 wt%, and 0.6 wt%) were dissolved in 3 mL deionized water. After stirring at 500 rpm for 30 min, 368 μL LMA was added into the above solution and continuously stirred for 3 h. Then, 2 mL AA and 2.5 g LCNF suspension (mass fraction: 2%) were slowly added into the above solution and stirred for 1 h at 700 rpm/min. Afterwards, 0.025 g APS and 100 μL TMEDA were added and mixed for 5 min. The resulting liquid was poured into the mold and the single-network hydrogels containing sodium alginate were obtained after polymerization at 80 °C for 3 h. Finally, the single-network hydrogel was impregnated in 4 wt% CaCl_2_ to obtain the double-network structure, and the DN hydrogels were respectively named DN-0.2% SA, DN-0.3% SA, DN-0.4% SA, DN-0.5% SA, and DN-0.6% SA hydrogels according to SA content. Taking the DN-0.2% SA as an example, the weight of the hydrogel was 8.14 g. The weight of AA, SA, LCNF, and CaCl_2_ was 2 g, 0.016 g, 0.05 g, and 2 × 10^−7^ g, respectively. The water content was 66.7%.

### 2.3. Methods

To analyze the internal microstructure of the DN hydrogel, the hydrogel was fractured in liquid nitrogen to expose its cross-section. The fractured cross-section morphology of the DN hydrogel was observed using a scanning electron microscope (SEM) (Hitachi Regulus SU-70, Tokyo, Japan) at an accelerating voltage of 5 kV. Additionally, Image J software (Version 1.48) was utilized for SEM image analysis to survey the pore size distribution of the fractured cross-section of the hydrogel.

The mechanical properties of the DN hydrogel were analyzed by TA.XT plus C tensile and compression analyzer (Stable Micro Systems, Godalming, UK). For tensile testing, the hydrogel samples were cut into rectangular shapes (10 mm length × 1 mm width× 10 mm height) using a knife, and the tensile speed was 40 mm/min at 25 °C. For compression testing, the hydrogel samples were molded into a cylinder shape (20 mm diameter × 10 mm height), and the test speed was 10 mm/min at 25 °C. For the cyclic compression testing, the speed was 100 mm/min, and the maximum strain was set to 80%. Each group of samples was tested three times.

The thermal stability of the DN hydrogels was characterized by a thermogravimetric analyzer (TGA Q50, TA, Newcastle, DE, USA). The temperature rose from 25 °C to 600 °C at a heating rate of 10 °C/min under the protection of dry nitrogen. The chemical structures of all samples were characterized using a Fourier transform infrared spectrometer (FTIR, ALPHA, Bruker, Ettlingen, Germany) in the range from 4000 to 500 cm^−1^.

A digital source meter (Type 2450, Keithley, Cleveland, OH, USA) was used to test the conductivity of the hydrogel. The DN hydrogel was cut into rectangular strips and connected to two electrode clamps and the resistance value was recorded. The conductivity (σ) was calculated by the following Equation (1):(1)σ=L/RS
where *L*, *R*, and *S* are the thickness, the resistance, and the electrode area of the DN hydrogel, respectively.

The electrocardiogram (ECG) and electromyography (EMG) signals were recorded using a multichannel physiological signal acquisition and process system (RM6240CD, Chengdu Instrument Factory, Chengdu, China). To record the ECG signal, the DN hydrogel slices were fixed on the arm and on the interior side of the ankle of the experimenter, and the experimenter was asked to remain stationary. For the EMG signal recording, the experimenter was asked to repeatedly clench and relax their fist regularly so that the muscle was in a regular switching state between tension and relaxation.

The samples used for SEM, FTIR, and thermal stability tests were dry hydrogels obtained through freeze-drying under −72 °C vacuum conditions. The samples used for stretching, compression, and ECG, EMG tests were wet hydrogels that underwent simple cutting after synthesis.

## 3. Results and Discussion

### 3.1. Design Strategy of the DN Hydrogel

With the rapid development of artificial intelligence, there is an increasing demand for flexible wearable devices, especially flexible strain sensors. Strain sensors can quickly and repeatedly generate electronic signal changes, such as resistance, current, or voltage, in response to external forces such as strain or pressure. As a result, strain sensors have gained growing attention in various applications such as electronic skin, bioelectrodes, and wearable devices. In recent years, several traditional approaches involving embedding conductive polymers into elastic substrates have been well developed for fabricating electronic devices. However, these devices suffer from poor flexibility, limited stretchability, and incompatibility with human skin due to their rigid nature, which severely restricts their application in flexible strain sensors. Therefore, the development of a novel hydrogel with integrated stretchable, conductive, and thermal stable properties is the goal of this study.

As shown in Figure 1, deionized water was used as the solvent, and an appropriate proportion of SA was added. SA is rich in carboxylic acid groups and, when dissolved in water, can interact with water molecules through hydrogen bonding and van der Waals forces, resulting in a loosely structured hydrogel state. Acrylic acid was then added while stirring to prevent rapid addition and subsequent flocculation. Finally, the dispersed suspension of LCNF was added to the aforementioned system. Under the action of an initiator, acrylic acid underwent free radical polymerization, forming long PAA chains. Crosslinking between PAA and LCNF served as a matrix network. Furthermore, after immersion in a CaCl_2_ solution, significant changes occurred in the initially loose sodium alginate hydrogel. Ca^2+^ ions replaced Na^+^ ions in the sodium alginate, leading to the formation of the Ca^2+^-SA network, which acted as a sacrificial network. Additionally, due to the presence of numerous hydrophilic groups, SA, PAA, and LCNF formed an extensive hydrogen bonding network, further enhancing the toughness of the hydrogel. Importantly, the introduction of SA-Ca^2+^ resulted in a conductive DN hydrogel, offering a new approach for the development of strain-flexible sensors.

### 3.2. Molecular Structures

The FTIR spectra of the samples are shown in Figure 2. Peaks at 1054 cm^−1^ and 1451 cm^−1^ are caused by asymmetric vibration of C–O–C in LCNF and bending vibration of C–H in alkyl of PAA, respectively. Due to the hydrogen bond interactions between PAA, LCNF, and SA, and the physically crosslinking interactions between SA and Ca^2+^, the vibrations of C=O and C–H in PAA are red-shifted from 1700 (PAA–LCNF) to 1707 cm^−1^, while the vibrations of O–H and C–O in PAA are blue-shifted from 1631 (PAA–LCNF) to 1626 cm^−1^. The wide absorption peak at around 3650~3100 cm^−1^ becomes wider with the increase of SA content, indicating that the hydrogen bonds and physically crosslinking interactions increased.

### 3.3. Thermal Properties

TG and DTG curves of the DN hydrogel are shown in Figure 3a,b, and the corresponding data of the onset temperatures of thermo-decomposition (*T*_onset_), temperatures at the maximum weight loss (*T*_max_), and residual carbon rate are listed in Table 1. Based on the TG curve presented in Figure 3a, the thermal degradation process of the DN hydrogel can be categorized into three distinct stages. The initial stage occurs from 25 °C to approximately 130 °C, wherein a small weight loss is attributed to the evaporation of absorbed water within the hydrogel. Subsequently, the second stage exhibits a significant weight loss between 250 °C and 470 °C, resulting from the rupture of the molecular main chains. During this stage, the hydrogel sample undergoes decomposition, forming smaller molecules and gaseous byproducts. Finally, the third stage commences above 470 °C, during which the remaining residues further decompose into gas and char residue. With the increase of SA content, the *T*_onset_ and *T*_max_ first increase and then decrease, while the residual rate of all samples decreases. These results indicate that introducing SA-Ca^2+^ networks into PAA–LCNF networks effectively enhances the thermal stability of PAA–LCNF hydrogels, possibly through enhancing the hydrogen bonding interactions among PAA, SA, and LCNF, and thus limiting the mobility and dissociation of PAA–LCNF networks. However, when the SA content is larger than 0.4%, the loose SA gel network leads to slight decrease in *T*_onset_ and *T*_max._

### 3.4. Morphology Analysis

The SEM images of the DN hydrogels are shown in Figure 4a. The DN hydrogels have a porous structure with intersecting pore channels. The diameter distributions of pores are illustrated in Figure 4b. With the increase of SA content, the pore size reduces, since the physically crosslinking interactions between SA and Ca^2+^ can increase the crosslinking density of hydrogel network and thus limits the swelling of the hydrogel [35]. However, the pores size increases when the SA content is over 0.5%. This is attributed to the fact that the sodium alginate forms a viscous liquid state when dissolved in water. When the concentration of sodium alginate exceeds a certain value, it reaches saturation in terms of crosslinking with other components. The excess sodium alginate spontaneously aggregates within the system, loosening the initial compact crosslinked structure to become loose, and reduces the pore density of the hydrogel [36].

### 3.5. Mechanical Properties

The mechanical properties of hydrogels are crucial to their applications, including strength, toughness, elasticity, and recoverability, as well as durability and stability. These properties determine whether the hydrogels can adapt to various environments and conditions, and maintain stable and effective performance. For applications such as wearable devices, biomedical devices, and tissue engineering, hydrogels need to have sufficient strength and toughness to withstand external stresses and stretching. Additionally, good elasticity and recoverability enable the hydrogels to adapt to external deformations and recover their original shape after deformation. Furthermore, hydrogels also need to remain stable and durable under various environmental conditions to resist the erosion of biological fluids and maintain their chemical and physical properties. Therefore, developing new polymer network structures and crosslinking systems, as well as exploring new processing methods and technologies, are essential to enhancing the mechanical properties and stability of hydrogels to meet application needs.

Herein, the tensile and compressive test of DN hydrogels were conducted and the results are shown in Figure 5 and Figure 6, respectively. Figure 5 depicts the tensile stress-strain curves of DN hydrogels with different SA contents. The maximum tensile strain of DN hydrogel is significantly higher than pure PAA–LCNF hydrogel. When the content of SA in DN hydrogel is 0.5 wt%, the elongation at break increases from 3466 to 5607%. The introduction of alginate can increase the crosslinking sites and thus make the network denser, resulting in the improved deformation capability of the hydrogel. Interestingly, the elongation at break of DN hydrogels reduces when the SA content further increases to 0.6 wt%. The possible reason is that excessive SA would make the gel networks fragile since the SA gel network is loose. Since the DN hydrogel with 0.4 wt% SA content has the optimal mechanical properties, the cyclic tensile test of this sample was conducted, and the results is shown in Figure 5d. Significant hysteresis is observed from the resulting curves at 300% strain. This hysteresis is attributed to the breaking of the physically crosslinked network of SA-Ca^2+^ and hydrogen bond. Compared with the test results of PAA–LCNF hydrogel (Figure 5e), the maximum stress of DN hydrogel is greater than that of PAA–LCNF hydrogel in each tensile cycle.

Figure 6 shows the results from the compression stress-strain test of DN hydrogel. The compressive stress at 80% strain rises as the SA content increases and reduces when the SA content is larger than 0.4%, showing a similar trend as the tensile test. This can also be attributed to the synergetic effect of the enhanced DN and the loose SA gel network. Additionally, a cyclic compression test of the DN hydrogels with 0.4% SA content was carried out and the results are shown in Figure 6d,e. During the unloading process, there exists irreversible deformation, especially in the unloading of the first cycle, and the maximum stress of each cycle slightly decreases. This reduction in mechanical properties can be attributed to the fracture of the relatively fragile SA-Ca^2+^ sacrificial network.

### 3.6. DN Hydrogel Sensor for Detecting Human Motions

Hydrogel, as a flexible material, is widely used in the field of wearable flexible strain sensing. It can monitor human movement signals such as steps and heart rate, which is of great significance for health management and sports training. When using hydrogel as a strain sensor, in addition to requiring excellent mechanical properties of the hydrogel itself, its conductivity is also extremely important. Hydrogel strain sensors usually monitor strain signals by measuring resistance changes. Therefore, good conductivity can ensure that the sensor can respond quickly and accurately when subjected to strain, and convert resistance changes into readable signals. The performance of conductivity also affects the sensitivity and resolution of the sensor. Hydrogels with excellent conductivity can reduce the strain required for resistance changes, resulting in significant resistance changes at smaller strains. This allows the sensor to be more sensitive to capture small strain changes, improve resolution, and more accurately monitor signals in human movement or biomedical applications. In addition, conductivity affects the propagation speed of current, and good conductivity can quickly transmit resistance signals when strain signals are generated. This enables real-time response and data transmission, meeting the needs of dynamic monitoring.

The conductivity of the DN hydrogels is illustrated in Figure 7a. The DN hydrogel demonstrates good conductivity, owing to the presence of Na^+^, Ca^2+^ and the –COO^−^ inside DN hydrogel networks. The conductivity of the DN hydrogel enables applications as flexible sensors to monitor the electrophysiological signals. The electromyography (EMG) and electrocardiogram (ECG) signals acquired from DN hydrogel sensors are shown in Figure 7b and Figure 7c, respectively. The EMG signals from the DN hydrogel sensor can reflect the different activities when the experimenter clenches or relaxes their fist. The signals in Figure 7c can also clearly show each stage of the ECG signal.

## 4. Conclusions

In this study, we developed a stretchable, electrically conductive, and thermally stable hydrogel by introducing SA into PAA–LCNF-based hydrogel. The DN hydrogel is a porous structure with double network which is formed through hydrogen bonding and physically crosslinking interactions between SA-Ca^2+^, PAA, and LCNF; the covalent network of PAA–LCNF acts as the resilient framework of the hydrogel, while the calcium bridging networks of SA, along with the robust hydrogen bonding network within the system, function as sacrificial bonds that dissipate energy and facilitate stress transfer. The introduction of SA can effectively enhance the mechanical property and conductivity of the DN hydrogel. Since the SA network is loose and fragile, the content of SA is vital to the properties of DN hydrogel, and the properties will decrease when the SA content is excessive. From the test results, the DN hydrogel with 4 wt% SA content demonstrated optimal performance with large elongation at break (5607%), good thermal stability (*T*_onset_ = 268.7 °C and *T*_max_ = 396.4 °C), and good electrical conductivity (3.22 × 10^−2^ s/m). The application of DN hydrogel as flexible sensors in biomedical measurement is also demonstrated. Our results can provide a reference for developing multifunctional hydrogels that can withstand ultra large deformation.

## Figures and Tables

**Figure 1 polymers-16-00102-f001:**
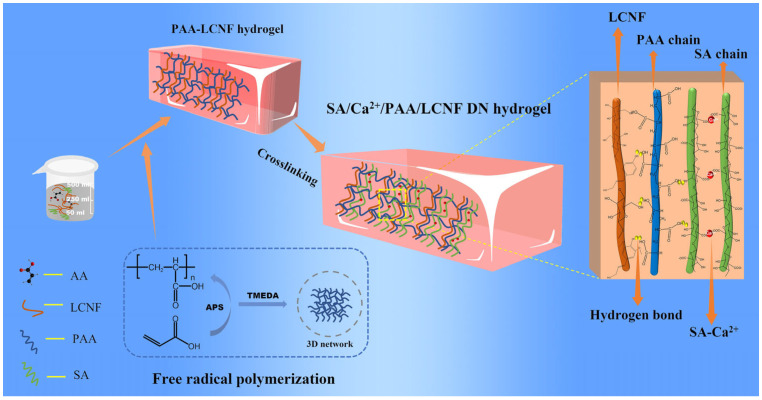
Fabrication process of the SA/Ca^2+^/PAA/LCNF double-network hydrogel.

**Figure 2 polymers-16-00102-f002:**
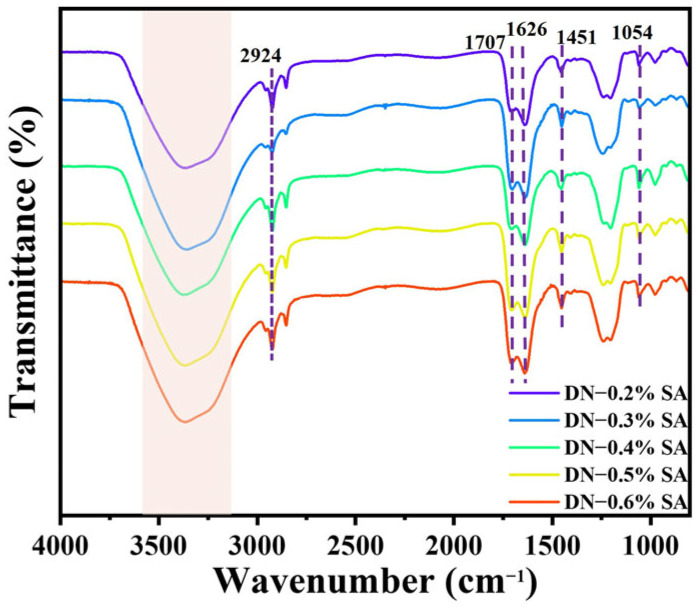
FTIR spectra of the DN hydrogel.

**Figure 3 polymers-16-00102-f003:**
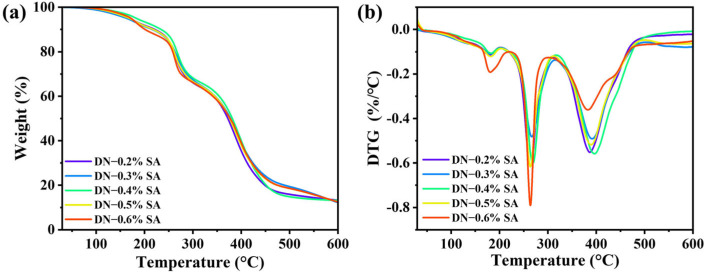
(**a**) TG and (**b**) DTG curves of the DN hydrogel.

**Figure 4 polymers-16-00102-f004:**
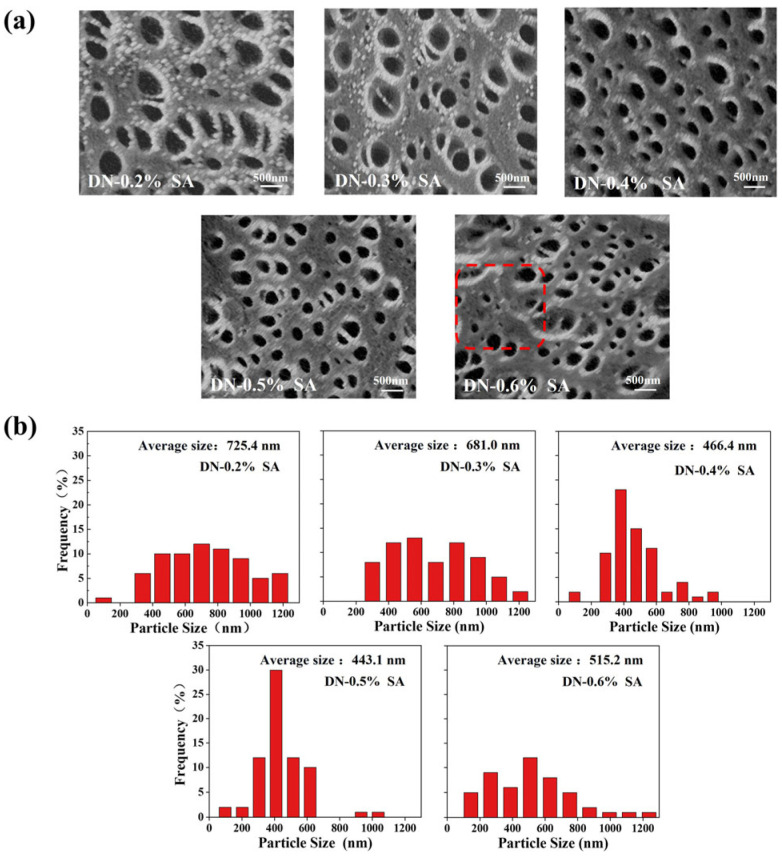
(**a**) Pore structures and (**b**) diameter distributions of pores in the DN hydrogel.

**Figure 5 polymers-16-00102-f005:**
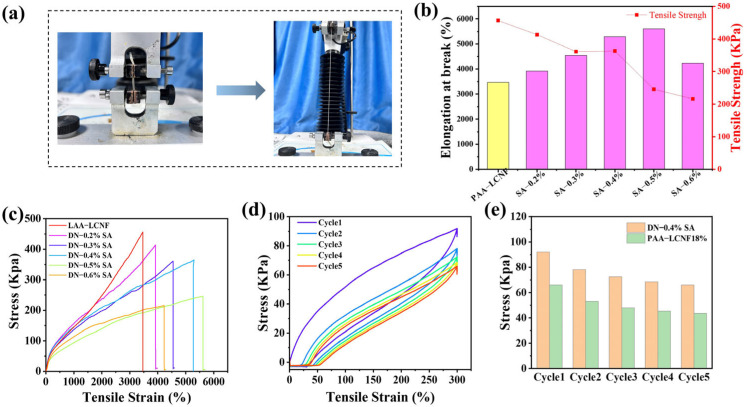
(**a**) Tensile stress-strain tests. (**b**) Tensile stress-strain curve. (**c**) Elongation at break and tensile strength. (**d**) Cyclic stretching of DN-0.4% SA hydrogel. (**e**) Tensile cycling of PAA–LCNF and DN hydrogel.

**Figure 6 polymers-16-00102-f006:**
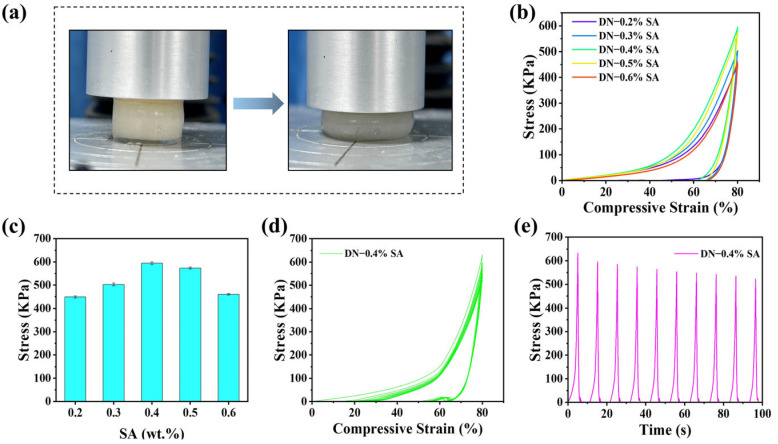
(**a**) Compressive stress-strain tests. (**b**) Compressive stress-strain curves. (**c**) Compressive strength of DN hydrogels with different SA contents. (**d**) Fatigue compressive stress–time curves of DN-0.4% SA hydrogel for a loading–unloading cycle at 80% strain without resting intervals. (**e**) Self-recovery and fatigue resistance of DN hydrogel (ten successive loading–unloading cycles in 100 s).

**Figure 7 polymers-16-00102-f007:**
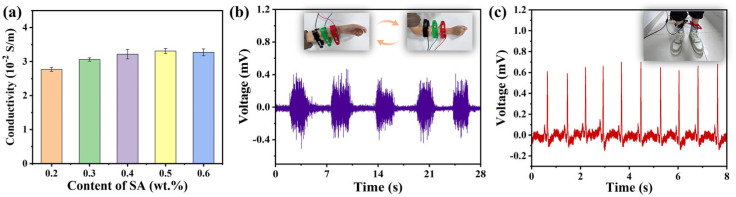
(**a**) Conductivity, (**b**) EMG, and (**c**) ECG signals detected by the DN hydrogel sensor.

**Table 1 polymers-16-00102-t001:** The values of *T*_onset_, *T*_max_, and residual rate of DN hydrogels in a temperature range 25–600 °C.

Sample	*T*_onset_ (°C)	*T*_max_ (°C)	Residual Rate (%)
DN-0.2% SA	263.9	386.7	13.2
DN-0.3% SA	266.3	392.2	13.2
DN-0.4% SA	268.7	396.4	12.2
DN-0.5% SA	263.7	389.8	12.5
DN-0.6% SA	263.9	383.4	12.4

## Data Availability

The data presented in this study are available on request from the corresponding authors.

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
