# Peer review of "Enhancing the Toughness of PAA/LCNF/SA Hydrogel through Double-Network Crosslinking for Strain Sensor Application"

_polymers, 2023, doi:10.3390/polym16010102_

Round 1

Reviewer 1 Report

Comments and Suggestions for Authors

The manuscript polymers-2765318 demonstrates the development of the method for obtaining of polyacrylic acid based hydrogels with lignin-containing nanocellulose fibers and in my opinion can be published in Polymers. However, I think that the manuscript can be improved for reading. This will increase interest for a wider range of readers. Here are some suggestions:

1. Abstract should show the essence of the article, in my opinion, and not be only a summary of what the authors did. It is not clear what the achievements of the authors are compared with the known data. What is fundamentally decided, what effect is achieved. Abstract must be corrected.

2. The title of the material does not correctly reflect the object of study. Not labeled sodium alginate.

3. Line 16: What is SA?

4. No actual data on the composition of DN hydrogels is provided: water content, Ca, actual molar ratio AA : SA.

5. The concept “double-network hydrogel” was not used correctly. There is not a single proof of the formation of the designated structure.

6. The “Discussion” section should be renamed “Conclusion”.

7. Conclusion needs to be improved. Conclusion should show the essence of the article, in my opinion, and not be only a summary of what the authors did. It is not clear what the achievements of the authors are compared with the known data. What is fundamentally decided, what effect is achieved.

Reviewer 2 Report

Comments and Suggestions for Authors

The authors have presented their studies on enhancing the toughness of PAA/LCNF hydrogels by employing double crosslinking. The paper falls within the scope of Polymers and can be considered for publication after attention to the following:

1. Dual crosslinking in hydrogels, with a combination of covalent and ionic bonds, has been widely reported in the literature. Hence, why is the current study novel? Please discuss the novelty in the introduction section.

2. Line 50, change "enhancing" to enhance.

3. Section 2.3. Please specify whether dry or wet hydrogels samples tested, if dry samples were tested then how were they dried?

4. Section 3 - heading should be changed to "Results and discussions".

5. Section 3.3. Please explain why excessive SA leads to looser networks and how this can lead to lower pore density?

6. Section 4 - heading should be "Conclusion", not Discussion.

Comments on the Quality of English Language

Quality of English language is acceptable. There are some minor corrections which have been listed above.

Round 2

Reviewer 2 Report

Comments and Suggestions for Authors

In my opinion, the authors have attended to the reviewer's comments satisfactorily. The manuscript has been improved significantly and it can be considered for publication now.